# Phase Transformations and Subsurface Changes in Three Dental Zirconia Grades after Sandblasting with Various Al_2_O_3_ Particle Sizes

**DOI:** 10.3390/ma14185321

**Published:** 2021-09-16

**Authors:** Hee-Kyung Kim, Kun-Woo Yoo, Seung-Joo Kim, Chang-Ho Jung

**Affiliations:** 1Department of Prosthodontics, Institute of Oral Health Science, Ajou University School of Medicine, Suwon 16499, Korea; 2Department of Chemistry and Department of Energy Systems Research, Ajou University, Suwon 16499, Korea; alex0227@ajou.ac.kr (K.-W.Y.); sjookim@ajou.ac.kr (S.-J.K.); 3Department of Mechanical Engineering, Ajou University, Suwon 16499, Korea; did9594@ajou.ac.kr

**Keywords:** air abrasion, zirconium oxide, dental stress analysis, phase transition, surface tension

## Abstract

Although sandblasting is mainly used to improve bonding between dental zirconia and resin cement, the details on the in-depth damages are limited. The aim of this study was to evaluate phase transformations and subsurface changes after sandblasting in three different dental zirconia (3, 4, and 5 mol% yttria-stabilized zirconia; 3Y-TZP, 4Y-PSZ, and 5Y-PSZ). Zirconia specimens (14.0 × 14.0 × 1.0 mm^3^) were sandblasted using different alumina particle sizes (25, 50, 90, 110, and 125 µm) under 0.2 MPa for 10 s/cm^2^. Phase transformations and residual stresses were investigated using X-ray diffraction and the Williamson-Hall method. Subsurface damages were evaluated with cross-sections by a focused ion beam. Stress field during sandblasting was simulated by the finite element method. The subsurface changes after sandblasting were the emergence of a rhombohedral phase, micro/macro cracks, and compressive/tensile stresses depending on the interactions between blasting particles and zirconia substrates. 3Y-TZP blasted with 110-µm particles induced the deepest transformed layer with the largest compressive stress. The cracks propagated parallel to the surface with larger particles, being located up to 4.5 µm under the surface in 4Y- or 5Y-PSZ subgroups. The recommended sandblasting particles were 110 µm for 3Y-TZP and 50 µm for 4Y-PSZ or 5Y-PSZ for compressive stress-induced phase transformations without significant subsurface damages.

## 1. Introduction

Zirconia ceramics doped with 3 mol% Y_2_O_3_ (3 mol% yttria-stabilized tetragonal zirconia polycrystal; 3Y-TZP) play a key role in dentistry since they are used in diverse important applications such as crowns, endodontic posts, orthodontic brackets and dental implants. In addition, improvements in the manufacturing process of zirconia ceramics combined with computer aided design/computer aided manufacturing (CAD/CAM) technologies are capable of creating the repeatable fabrication of individualized dental prosthesis with high accuracy [1]. One of the main reasons for the considerable interest in the dental community would be their highest mechanical strength among ceramic oxides. Unlike other ceramics, zirconia is a metastable ceramic, consisting of monoclinic, tetragonal, and cubic phases depending on the temperature. The superior mechanical properties of zirconia ceramics are attributed to the stress-induced transformation toughening mechanism around a crack tip [2].

Although zirconia ceramics exhibited superior fracture toughness and strength, their inherent opacity often cannot satisfy patients’ esthetic demands. With higher yttria contents (4 or 5 mol% partially stabilized zirconia; 4Y-PSZ or 5Y-PSZ), the amount of isotropic cubic phase increases and thus, a significant enhancement in translucency has been obtained due to the reduced light scattering at the grain boundaries [3]. Since the tetragonal to monoclinic phase transformation under tension is the main factor to determine the fracture toughness, the mechanical strengths of highly translucent zirconia would be compromised due to the limited amount of metastable tetragonal phase [3].

With improved technology, zirconia ceramics are now the preferred dental restorative materials, replacing metal-based restorations. However, in terms of clinical durability, one of the major concerns associated with zirconia restorations would be their problematic bonding to resin cements due to their chemically inert nature and hardness [1]. In particular, abrasive blasting, more commonly known as sandblasting, has been implemented to roughen the surfaces of hard and brittle zirconia for better adhesion. Previous studies have evaluated the effect of sandblasting on the bonding efficiency [4,5,6,7], surface topographies [4,7,8,9], mechanical properties [7,10,11,12,13,14,15], and phase transformations [4,5,6,7,8,10,11] of dental zirconia. Those studies included alumina abrasive particles (grain size: 25–125 µm) and compressed air (pressure: 0.1–0.4 MPa) which are mixed to form high-speed abrasive flow after passing through the nozzle. Herein, the kinetic energy formed from the abrasive mass and velocity would impact the zirconia surfaces creating micro-removal finishing [9]. When a brittle material is impacted by a hard particle, plastic deformation occurs in front of a crack tip creating compressive and shear stresses and as a result, the radial cracks are generated directly under the impact zone [16]. After yield point, tensile residual stresses left in an material on unloading would cause lateral cracks provoking the material removal [16]. Furthermore, the properties of the sandblasting particles, such as hardness, size, shape can affect their kinetic energy as a result of the particle-target interactions [8]. The kinetic energy of the blasting particles decreased by 3.5 times as the particle’s diameter decreased from 670 µm to 420 µm, reducing the plastic deformation of the surface [17]. Thus, the impact of hit can depend on the particle mass. Essentially, the bigger blasting particles caused morphological defects in the zirconia surfaces [8].

Several studies demonstrated that sandblasting triggered the phase transformation and improved flexural strength of 3Y-TZP [12,13,14], while severe sandblasting conditions might deteriorate the mechanical strength of 3Y-TZP yielding excessive monoclinic contents [15]. Chintapalli et al. [15] suggested that low impact angle of 30° should be applied when sandblasting with larger alumina particles (110 µm) for 3Y-TZP to decrease the adverse effects. Wongkamhaeng et al. [14] reported that air-abrasion on the zirconia surface (3Y-TZP) with the coarser alumina grains (250 µm) produced the larger subsurface damages; higher monoclinic phase value (11.2 ± 0.4%) and subsurface damages to a depth of 40.3 ± 20.3 µm. 

Recently, the adhesive bonding behaviors of highly translucent monolithic dental zirconia were investigated. It was shown that highly translucent zirconia (5Y-PSZ) revealed lower bond strength to resin cements compared with conventional zirconia (3Y-TZP), although mechanical sandblasting significantly improved bond strength of highly translucent zirconia [5]. Inokoshi et al. [11] evaluated the effect of alumina air-abrasion on the flexural strength of highly translucent zirconia materials and revealed that sandblasting increased or decreased their flexural strengths depending on the formation of microcracks and surface compressive stresses. Alumina sandblasting significantly increased surface roughness values of all zirconia grades and the amount of residual stresses generated by sandblasting were dependent on the substrate materials and alumina particle sizes [8]. Zhao et al. [18] suggested that less kinetic energy of the impinging particles would be required to get an adequate degree of roughening for 5Y-PSZ compared to that for 3Y-TZP.

Since a cubic phase does not undergo the stress-induced phase transformation, the presence of a cubic structure in highly translucent zirconia is responsible for a significant reduction in the mechanical properties [1]. The highly translucent zirconia materials exhibit different chemical compositions and microstructures compared to conventional zirconia [1] and thus, they might behave differently from conventional counterparts in the sandblasting processes. When considering the sandblasting process to improve the bonding efficiency of highly translucent zirconia, special attention might be paid to avoid possible surface damages. However, an optimal protocol for the sandblasting parameters of highly translucent zirconia has not yet been established. It was reported that there was a gradient in monoclinic phase up to a depth of about 13 µm and the highly deformed thin layer contributed to the residual stresses in the sandblasted 3Y-TZP ceramics [19]. Zhang et al. [16] reported that the radial crack generated below the plastic deformation zone decreased the strength of zirconia materials, while lateral crack caused the material removal. With regard to newly introduced highly translucent zirconia, there is currently no details on the subsurface changes after sandblasting. 

Therefore, the purpose of this study was to evaluate the phase transformations and subsurface changes after sandblasting with various Al_2_O_3_ particles in three different dental zirconia (3Y-TZP, 4Y-PSZ, and 5Y-PSZ), which would be crucial to predict the long-term mechanical behavior of dental zirconia systems. In this study, the subsurface changes of phase transformations and the crystal strain/stress state after sandblasting were quantified by using X-ray diffraction (XRD) and Rietveld analysis. The extent of subsurface alterations was determined with focused ion beam nanotomography (FIB-nt) through a non-destructive, serial slicing procedure allowing the observation of first few microns below the surface. In this study, five different sizes of commercially available Al_2_O_3_ blasting particles were used to verify the effect of different kinetic energy (alumina particle size) on the subsurface changes in three different dental zirconia substrates. To provide in-depth residual stress distribution within the zirconia components after being subjected to sandblasting, 3D-finite element analysis (FEA) was performed. The null hypothesis tested in this study was that there would be no difference in the subsurface changes after sandblasting with different alumina particle sizes between three different zirconia grades.

## 2. Materials and Methods

### 2.1. Specimen Preparation

Three different grades of commercially available dental zirconia materials were investigated, including one conventional tetragonal zirconia (3Y-TZP; KATANA ML, Kuraray Noritake Dental, Tokyo, Japan) and the new generations of highly translucent cubic-phase-containing zirconia (4Y-PSZ and 5Y-PSZ; KATANA STML and KATANA UTML respectively, Kuraray Noritake Dental). Fully sintered plate-shaped zirconia specimens of each grade (14.0 mm × 14.0 mm × 1.0 mm) were polished with 400, 600, and 800-grit silicon carbide papers and thermally etched in air at 1400 °C for 30 min. Then, specimens of each grade (n = 12 per each zirconia grade) were divided into 6 groups according to the alumina abrasive particle size. Air abrasion was performed with five different sizes of alumina particles (25, 50, 90, 110, and 125 µm; Cobra, Renfert GmbH, Hilzingen, Germany) at a pressure of 0.2 MPa and a distance of 10 mm from the specimen’s surface for 10 s/cm^2^ using a sandblasting device (Basic master, Renfert). The specimen group of each grade which did not receive sandblasting served as the control. Only the polished surface was sandblasted. The sandblasting process is schematically illustrated in Figure 1. 

### 2.2. Phase Transformation and Compressive Strain Evaluation

One specimen from each experimental subgroup in each zirconia grade was submitted to determine the crystal structures and phase transformations. Powder XRD measurements were carried out using a DMAX-2200PC X-ray diffractometer (Rigaku, Tokyo, Japan) equipped with a graphite monochromator (λ_CuKα_ = 0.15418 nm). A step scan mode was employed in the 2θ range of 20–90° with a step size of 0.02° and counting time of 4 s for each step. Quantitative phase analysis was conducted with the Rietveld refinement method using the Fullprof program [20] and the intensity profiles were fitted by using pseudo-Voigt functions. 

The surface strain/stress induced by sandblasting was evaluated with the Williamson-Hall (W-H) method in the uniform deformation model (UDM) [21]. The physical broadening of XRD peak as a function of microstrain was considered according to Equation (1) [21,22]:(1)$${ß}_{\mathrm{hkl}}\cdot \cos\;\theta =(\frac{K\lambda }{L})+(4\varepsilon \cdot \sin\;\theta )$$
where L is the nanocrystal size; K is the shape factor, usually taken as 0.89 for ceramic materials; λ is the wavelength of radiation in nanometer; ß is the full width at half maximum of the peak in radians; θ is the diffracted angle of the peak; ε is the strain induced by crystal deformation.

### 2.3. Microstructural Analyses

The other specimen from each experimental subgroup in each zirconia grade was submitted to observe the microstructural changes in the near-surface zone during the sandblasting process. A thin layer of platinum (1 µm) was deposited on the specimen and then the cross-section specimens were prepared by using focused ion beam (FIB; ZEISS CrossBeam 540, Carl Zeiss Microscopy GmbH, Oberkochen, Germany) milling, equipped with a Zeiss Capella FIB column and a Gemini II SEM column. The sandblasted surface was milled with Ga^+^ ions at 30 kV by using a decreasing sequence of the ion currents, down to a final polishing step of 300 pA. The FIB/SEM images of each cross-section were obtained with the energy-selective backscatter (EsB) detector. The transformed zone depth on each FIB cross-section was measured for 10 randomly selected sites using ImageJ software (v1.53e, National Institutes of Health, Bethesda, MD, USA) with the line measuring tool to measure a length distance over an image.

### 2.4. D-Finite Element Analysis (FEA) 

The stress field subjected to a single blasting particle was simulated by the finite element method (FEM) with LS-DYNA software (v10.0, Livermore Software Technology Corporation (LSTC), Livermore, CA, USA). The material elastic properties of 3D-FEA models were shown in Table 1. After constructing the models, a linear elastic analysis under a dynamic load was performed. The indentation models assumed that the affected stress fields underneath the particle impact were hemispherical symmetries. Since ceramic materials could fail by brittle fracture, the maximum principal stress (MPS) was considered to evaluate the stress filed on loading below the impact zone. The analyses of tensile (positive) or compressive (negative) stresses at impact were conducted in the zirconia components of all models [23]. The contact between abrasive particle and the zirconia substrate was expressed by a simplified model of point-indentation microfracture patterns (Figure 2) as a function of alumina particle size. The region of material erosion was calculated by adaptive mesh and element deletion [23]. The meshing process was performed on the models with 10-node quadratic tetrahedral elements and the mesh size was set to 0.001 mm. According to the published data by Jafar et al. [24], the impact velocity of single abrasive particle can be estimated (Figure 3). The incident velocity of the impact particle was calculated by Equation (2):V_x_ = −46.76ln(x) + 307.77 (R^2^ = 0.98)(2)
where V is the velocity in m/s; x is the particle size in µm: V_25_ = 157 m/s; V_50_ = 125 m/s; V_90_ = 97 m/s; V_110_ = 88 m/s; and V_125_ = 82 m/s. The finite element models used in this study were based on the following assumptions: all materials were homogeneous with a linear elastic behavior under stress and there were no flaws within any component. 

### 2.5. Statistics

Statistical analyses were conducted using a software (IBM SPSS Statistics for Windows, v25.0, IBM Corp., Chicago, IL, USA) with a significance level of α = 0.05. Shapiro–Wilk test was performed to evaluate the normal distributions and Levene test was applied to verify the homogeneity of variance. Means of the transformed zone depth along the crystal structures were compared among experimental subgroups. A two-way analysis of variance (ANOVA) was applied to analyze the effect of the zirconia grade and the abrasive particle size on the transformed zone depth after sandblasting.

## 3. Results

### 3.1. XRD Analysis

Figure 4 reveals the XRD patterns and the enlarged graphs in the 2θ angle range from 27.5° to 30.5° and in the 2θ angle range from 58.5° to 60.5° for all subgroups of each zirconia grade: (a) and (b) for 3Y subgroups; (c) and (d) for 4Y subgroups; and (e) and (f) for 5Y subgroups. Figure 5 demonstrates quantitative phase distributions obtained from Rietveld refinements for (a) 3Y subgroups, (b) 4Y subgroups, and (c) 5Y subgroups. As shown in Figure 4 and Figure 5, the control groups generally had two crystal structures (tetragonal and cubic phases) and the cubic contents increased with increasing Y_2_O_3_ content (32.9 wt% for 3Ycon, 51.4 wt% for 4Ycon, and 53.7 wt% for 5Ycon). The amount of monoclinic phase is considered to be negligible in the control groups. After sandblasting, the specimens of each grade displayed the asymmetrical peak broadening of (011)_t_ at 2θ = 30.27° and (112)_t_ at 2θ = 50.38°, which became broader as the bigger blasting particle was used. As shown in Figure 4, sandblasting caused the (011)_t_ peak shift to a higher angle and the maximum peak shift occurred in 3Y110 for 3Y subgroups, in 4Y110 for 4Y subgroups, and in 5Y25 for 5Y subgroups. Generally, the smallest peak shift was observed in 5Y subgroups among three zirconia grades.

Sandblasting conditions used in this study produced only a small fraction of monoclinic phases (0–2.3 wt%), with highest value of 2.3 wt% in 3Y125 for 3Y subgroups, 1.8 wt% in 4Y25 for 4Y subgroups, and 2.18 wt% in 5Y50 for 5Y subgroups, being zero for 4Y50, 4Y90, 4Y110, 4Y125, 5Y90, 5Y110, and 5Y125. After sandblasting, the rhombohedral phase (r-ZrO_2_) [27] was identified on the low angle side of the (011)_t_ peak for all zirconia grades with a maximum intensity (12 − 1)_r_ peak at 2θ = 29.88. Figure 5 demonstrates that the tetragonal phase contents decreased while the rhombohedral phase contents increased (up to 64.38 wt% for 4Y subgroups and up to 57.01 wt% for 5Y subgroups) as the blasting particle size increased for all zirconia grades. For 5Y subgroups, the t-phase content rapidly reduced to zero in 5Y90.

The strain associated with the sandblasting of three zirconia grades using different sizes of Al_2_O_3_ was compared and plotted from the interplanar spacing by a modified form of W-H analysis, UDM, based on the assumption of isotropic nature of the crystal under stress (Figure 6). As shown in Figure 4, the intensity of tetragonal (011) peak at 2θ = 30.27° was sharp and narrow in control groups, confirming that the polished specimen (control) of each zirconia grade exhibited a high degree of crystallinity. The strain-induced broadening of the tetragonal (011) peak due to lattice deformation [28] after sandblasting was calculated and a plot was drawn with 4 sin θ along the *x*-axis and β cos θ along the *y*-axis as depicted in Figure 6. The linear function fitted to the experimental data for each subgroup and the strain/stress was evaluated from the slope values. A plot of 3Y110 showed the steepest negative slope, indicating the largest value of compressive strain/stress. The positive slope values would be related to the tensile strain/stress while the negative values would be related to the compressive strain/stress [21,29]. For 4Y or 5Y subgroups, there were no lines with negative slopes. For 4Y subgroups, there were no further changes in the slope beyond 4Y50. For 5Y subgroups, severe peak broadening with low accuracy of the profile fitting was observed beyond 5Y50.

### 3.2. FIB/SEM Analysis 

Cross-sectional FIB-SEM images on the sandblasted specimens are shown in Figure 7, Figure 8 and Figure 9. The transformed zones with faceted grains were observed within the top few micrometers beneath the surfaces. These faceted crystals are considered to be the monoclinic grains occurred during the tetragonal-to-monoclinic transformation by a martensitic twinning mechanism [30] (shown with blue arrows in Figure 7e). In 3Y125, a small grain boundary microcrack was detected which was not interconnected. 

In contrast to 3Y-TZP, the cracks were interconnected and those intergranular or transgranular cracks could facilitate the zirconia material removal during the sandblasting [31] as the particle size increased in 4Y or 5Y subgroups. The propagation of microcracks was oriented parallel to the surface and those lateral cracks were located up to 4.5 µm distance under the surface in 4Y or 5Y subgroups. The transverse extension of cracks could generate the material removal, resulting in the brittle fracture. Compared to 4Y-PSZ, larger crack was induced with smaller blasting particle in 5Y-PSZ. No isolated transformed zone, but rather homogeneous defective layers were found under the surfaces of 4Y110 for 4Y subgroups and of 5Y90 for 5Y subgroups. Abnormal grain growth was observed in 5Y125, which may compromise the mechanical stability [32]. 

Figure 10 shows the extent of t-m phase transformation along the depth direction in all zirconia subgroups. A two-way ANOVA was conducted to examine the effect of zirconia grade and the abrasive particle size on the transformed zone depth. There was a statistically significant interaction between the effects of zirconia grade and the abrasive particle size on the transformed zone depth (*p* < 0.05). For 3Y subgroups, 3Y110 induced the deepest transformed layer up to 2.9 µm. For 4Y subgroups, 4Y25 induced the deepest transformed layer up to 0.8 µm. For 5Y subgroups, 5Y25 induced the deepest transformed layer up to 0.7 µm. However, there were no transformed layer detected beyond 4Y110 for 4Y subgroups and beyond 5Y90 for 5Y subgroups. 

### 3.3. Interaction between the Abrasive Particle and Zirconia Substrate

The hemispherical maximal principal stress fields on loading within the representative models of 3Y125, 4Y125, and 5Y125 are depicted in Figure 11. The compressive stress (negative) was concentrated at the impact zone and tensile stress (positive) was released approaching the surface. The affected zone near the blasted surface was divided into four zones: (i) plastic deformation zone with a depth of a few micrometers with a certain degree of microstructural deformations below the material erosion; (ii) tensile stress zone in which the tensile stress was generated at the surface; (iii) compressive stress zone in which the residual compressive stress was induced beneath the particle impact; (iv) stress relaxation zone in which the residual stress was partially released. 5Y125 model produced deeper erosive cut compared to 3Y125 or 4Y125 model with a depth of 2 µm. Below the material erosion, plastic deformation zone was developed. 5Y125 model exhibited deeper stress relaxation zone with a depth of 19 µm compared to 3Y125 or 4Y125. The maximum principal tensile/compressive stresses introduced to the model designs are presented in Table 2. The results demonstrated that the finite element models of 5Y subgroups exhibited lower values of MPS compared to the models of 3Y or 4Y subgroups.

Thicknesses of the affected stress layers and the material cuts below the impact zones for the models of all zirconia grades are shown in Figure 12. The depth of material erosion at the particle impact zone did not exceed 1 µm for the models of 3Y or 4Y subgroups, while it reached the maximum value of 2 µm for the models of 5Y subgroups. Larger particles induced deeper affected stress layers for all zirconia grades. The models of 5Y subgroups, due to their inferior mechanical properties, exhibited deeper stress dissipation compared to the models of 3Y or 4Y subgroups.

## 4. Discussion

This study evaluated the microstructural and crystallographic subsurface alterations and residual stresses caused by Al_2_O_3_ sandblasting with five different particle sizes in three different grades of dental zirconia. The subsurface changes induced by sandblasting in this study could be explained by the emergence of a new phase (rhombohedral), by the presence of microcrack, crack propagation, or material removal, and by the formation of compressive/tensile stresses depending on the interaction between the blasting media and the substrate surface. The sandblasting effects with different alumina particle sizes varied between different zirconia grades due to the different phase compositions and microstructures. Therefore, the null hypothesis was rejected.

Previous dental research [33,34,35,36] on zirconia ceramics identified the presence of the rhombohedral zirconia phase, which was first suggested by Hasegawa in 1983 [33], under mechanical stresses. Our XRD results agreed with a previous study [36], where alumina sandblasting induced the asymmetrical peak broadening and led to the formation of rhombohedral phase in all zirconia grades. In this study, the amount of rhombohedral phase increased (up to 64.38 wt%), while the amount of tetragonal phase decreased as the alumina particle size increased in all zirconia grades. From the results of Rietveld analysis, those rhombohedral phases seemed to derive from tetragonal as well as from cubic phase since larger amount of rhombohedral phase was identified in highly translucent zirconia. However, only a very small amount of monoclinic phase was formed after sandblasting in all zirconia grades. For 3Y subgroups, the amount of monoclinic phase slightly increased (up to 2.3 wt%) after sandblasting as the alumina particle size increased. However, the occurrence of monoclinic phase was hardly observed in 4Y50, 4Y90, 4Y110, and 4Y125 for 4Y subgroups and in 5Y25, 5Y90, 5Y110, and 5Y125 for 5Y subgroups. The absence of the monoclinic phase may be attributed to the limited amount of the tetragonal phase and to the increased amount of the rhombohedral phase. It was suggested that the rhombohedral phase can act as a barrier to further tetragonal to monoclinic phase transformation [37].

With regard to the increase or decrease in the strength of zirconia after sandblasting, Inokoshi et al. [11] reported that the balance between the induced compressive stress and microcrack formation should be considered. It was reported that the occurrence of a rhombohedral phase after grinding of 3Y-TZP ceramics has led to the subsurface damages due to an increase in volume with grain pull-out [35]. A recent study suggested that weakened mechanical properties of sandblasted highly translucent zirconia could be attributed to the formation of rhombohedral phase [27]. In this study, FIB cross-sectional images indicated that there was a monoclinic phase gradient up to a depth of 2.9 µm for 3Y subgroups. The 3Y110 induced the deepest transformed layer without microcrack formations and least amount of rhombohedral phase among 3Y subgroups, showing the largest compressive residual stress in the Williamson-Hall plots.

As shown in the FEA model in this study, a tensile stress developed along the surface on loading although the compressive residual stress induced by t-m transformation might act as the primary driving force against lateral cracking, contributing to the toughening mechanism. A previous study [38] demonstrated that only a small amount of monoclinic transformation with a thickness of several microns could increase the fracture toughness and biaxial flexural strength. In another previous study [39], the monoclinic zirconia content in the sandblasted 3Y-TZP increased up to 9.5%, leading to an increase in the flexural strength. In this study a small subsurface microcrack was observed in 3Y125, reflecting the concentration of high stress at zirconia grain boundaries. However, the subsurface tensile component might well be sufficiently large to suffer a ductile-to-brittle transition of material removal mode in the blasting process [40]. When a critical tension accumulation was reached to initiate fracture within zirconia materials, there would be a collapse of the grain and an increase in the material defect, resulting in the reduced flexural strength [41]. With a particle size below 110 µm, the plastic deformation mechanisms could be activated and the propagation of the cracks was suppressed. Ho et al. [42] reported that the compressive residual stress introduced by abrasive grinding on the zirconia surface increased with increasing the mechanical strength as long as the grinding process was carefully controlled not to increase the flaw size.

With the highly translucent zirconia ceramics used in this study, the transformed monoclinic symmetry during sandblasting was formed in the very surface layer: maximum value of 0.83 µm in 4Y50 for 4Y subgroups and maximum value of 0.77 µm in 5Y25 for 5Y subgroups. As the alumina particle size increased, sandblasting did not induce monoclinic transformed layers of isolated grains but rather amorphous transformed layers in highly translucent zirconia. The connecting cracks were developed between pores and ran to the surface to cause material removal with larger alumina particles, which might be detrimental to mechanical properties. The amount of rhombohedral phase tended to increase up to 64.38 wt% for 4Y subgroups and up to 57.01 wt% for 5Y subgroups. The detection of larger amount of rhombohedral phase in 4Y subgroups might be attributed to the presence of transformable tetragonal phase. Whether the rhombohedral phase was derived from the cubic phase or the tetragonal phase, larger amount of transformed rhombohedral phase in 4Y subgroups would be a result of a t→r transformation.

The Williamson-Hall analysis reported in this study revealed that no compressive lattice strain was induced during sandblasting in highly translucent zirconia. In 5Y subgroups, severe peak broadening without periodicity of crystallinity generated scattering profiles due to the existence of polycrystalline aggregates [28] as the particle size increased. Those amorphous transformed layers were also confirmed in the FIB images. It was reported that increased yttria content led to the reduced crack propagation resistance for dental zirconia [43]. In this study, 5Y subgroups were more susceptible to subsurface damages, such as subsurface lateral cracks, and plastic deformations, and surface melting. The surface melting and the abnormal grain growth found in 5Y125 may be due to the localized high temperature. During sandblasting, particle’s kinetic energy would transform to thermal energy, causing local melting of the surfaces of dental zirconia ceramics [9]. It was reported that grain pull-out and surface degradation of zirconia were promoted at high temperature [44] and the increased grain size could promote crack formation [45].

We performed sandblasting on the zirconia surfaces as a mechanical test to identify the crystallographic and morphological changes after sandblasting in three different zirconia grades. Additionally, the numerical analysis by means of the finite element method was used to determine theoretical stress distributions as well as to understand the empirical data from the mechanical testing. FEA models of 5Y subgroups in this study revealed deeper erosive cut and deeper affected stress layers. In addition, 5Y subgroups exhibited less amount of maximum principal stress than 3Y or 4Y subgroups, meaning decreased load-to-failure [46]. It was reported that soft substrate absorbed a large amount of kinetic energy under impact load, resulting in large plastic deformation [47]. The FIB cross-sectional images of highly translucent zirconia with larger particles in this study revealed that abrasive particle intruded into the zirconia substrate in the form of brittle fracture due to the interactions between alumina particles and zirconia materials. Larger particles could gain higher kinetic energy than smaller ones, being more susceptible to fracture [47]. Therefore, the lower damage tolerance of highly translucent zirconia compared to conventional zirconia should be taken into consideration when setting the sandblasting protocols.

Although sandblasting on highly translucent zirconia might deteriorate the flexural strength, the bonding ability to resin cement could be enhanced due to the microcracks created during sandblasting and thus, higher shear bond strength could compensate for the negative impacts on the mechanical properties [27]. With regard to tooth bonding, it was suggested that enamel microcrack healing with adhesive resin could prevent crack propagation, leading to increased fracture toughness [48]. A recent study [8] reported that with 25- µm alumina particles, there were no significant changes in Sa (arithmetic mean height) values for all dental zirconia grades. With 50-µm or bigger alumina particles, surface roughness values increased as the particle size increased (up to 0.76 ± 0.12 µm in Sa value) in all zirconia grades. However, Raman spectroscopy demonstrated that the larger particles could reduce residual compressive stresses for highly translucent zirconia.

The depth of the monoclinic transformation zone may strongly affect the amount of compressive residual stresses. In this study, for 3Y subgroups, larger particles increased the compressive residual stresses as determined by Williamson-Hall analysis, creating the transformation zone down to a depth of 1.1–2.9 µm beneath the surface. In contrast to 3Y subgroups, the transformed zone depth was less than 1 µm and thus, sandblasting did not induce any compressive stresses at the surface. The thickness of the transformed layer under externally applied stress might be too shallow to induce compressive stresses. The surface microcrack may be beneficial to enhance the bonding efficiency of dental zirconia by the infiltration of resin cements through the cracks, leading to inhibition of crack propagation. Thus, the microcrack sealing mechanism might contribute to an increase in the mechanical strength of zirconia materials. However, the presence of lateral cracks with surface connection may adversely affect the mechanical behavior of zirconia systems as well as the long-term reliability of the dental prosthesis. In this study, with larger alumina particles, the transverse cracking became wide at the several micrometers below the surface but was rarely deep enough to affect structural integrity in highly translucent zirconia.

In summary, dental zirconia ceramics of three different grades showed different sandblasting reactions to the different alumina particle sizes, which was attributed to their unique crystallographic and mechanical properties. Considering the effect of alumina particle sizes on the potential subsurface damages and the induced compressive stresses in three zirconia grades, the recommended alumina particle sizes would be 110 µm for 3Y-TZP and 50 µm for 4Y-PSZ or 5Y-PSZ for better bonding without a significant reduction in the mechanical strength. However, there are some limitations to these recommendations. In this study, sandblasting parameters such as pressure, duration, angle, and distance were fixed except the particle size. The effect of various parameters should be considered to control the kinetic energy of blasting particles. For highly translucent zirconia, air abrasion with other abrasive particles of low hardness rather than alumina to reduce the particle’s kinetic energy, chemical etching, or abrasive waterjet to reduce possible thermal damages could be considered in further study. In addition, the determination of mechanical properties and shear bond strength after sandblasting were not performed here. Further study should include the effect of sandblasting on mechanical properties and bond strength of dental zirconia.

## 5. Conclusions

We have evaluated the crystallographic and microstructural subsurface changes after alumina sandblasting with five different particle sizes in three different dental zirconia grades. Within the limitations of this in vitro study, the following conclusions were drawn:(1)Although alumina sandblasting induced tetragonal to monoclinic phase transformation in the conventional zirconia, the phase transformation would depend on the amount of metastable tetragonal phase in highly translucent zirconia.(2)When selecting the appropriate sandblasting protocols of highly translucent zirconia, care should be taken in order not to deteriorate the mechanical properties due to the high susceptibility to surface damage under applied stress.(3)Within the range of treatment parameters investigated in this study, the recommended sandblasting particles were 110 µm for 3Y-TZP and 50 µm for 4Y-PSZ or 5Y-PSZ at a pressure of 0.2 MPa and a distance of 10 mm from the specimen’s surface for 10 s/cm^2^ to implement compressive stress-induced phase transformations without significant subsurface damages.

## Figures and Tables

**Figure 1 materials-14-05321-f001:**
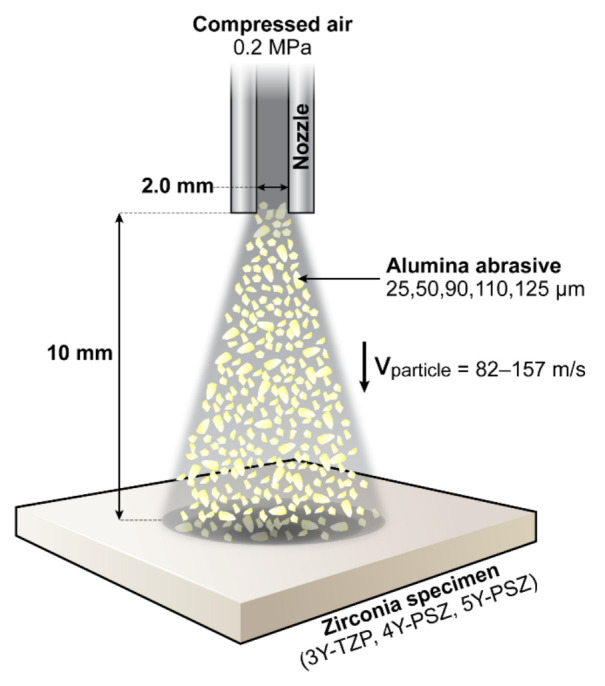
Schematic diagram of the sandblasting process; The alumina particles are accelerated towards the specimen with a high-pressure airflow through a circular nozzle (with a diameter of 2.0 mm). The alumina particles hit the specimen with a speed up to 157 m/s. 3Y-TZP: 3 mol% yttria-stabilized tetragonal zirconia polycrystal; 4Y-PSZ: 4 mol% partially-stabilized zirconia; 5Y-PSZ: 5 mol% partially-stabilized zirconia.

**Figure 2 materials-14-05321-f002:**
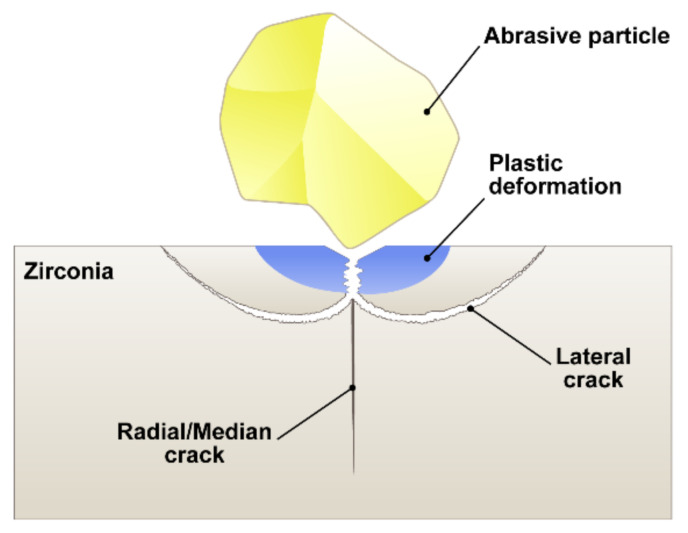
Model of point-indentation microfracture patterns [16]. Just below the particle impact, the plastic deformation zone can be generated. The radial crack and lateral crack start from the plastic zone.

**Figure 3 materials-14-05321-f003:**
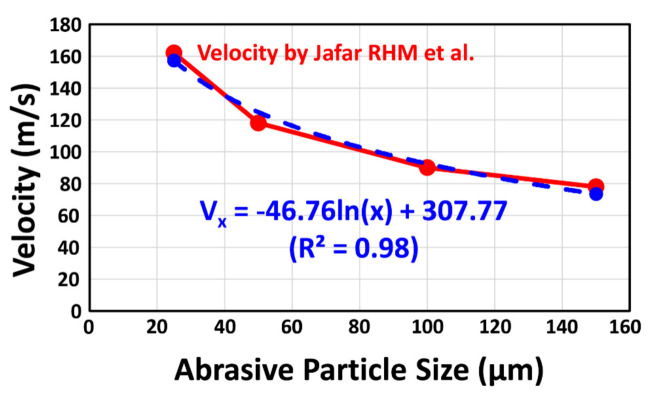
The impact velocity of single abrasive particle can be estimated according to the particle velocities measured by Jafar et al. [24] V: velocity (m/s); x: particle size (µm).

**Figure 4 materials-14-05321-f004:**
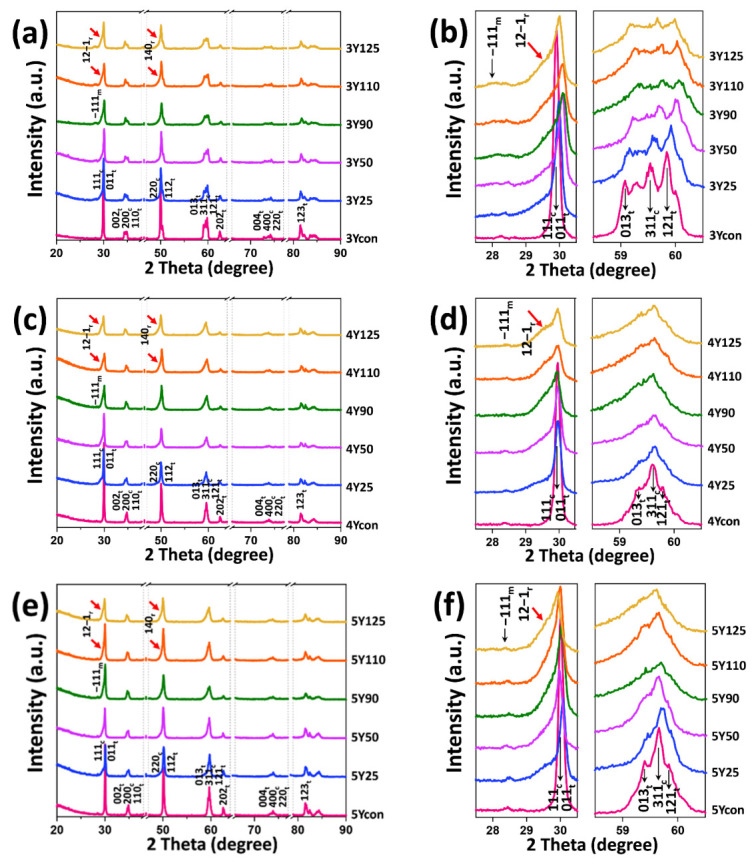
X-ray diffraction (XRD) patterns of subgroups subjected to sandblasting with different size of alumina particles for 3 mol% yttria-stabilized tetragonal zirconia polycrystal: (**a**) in the 2θ range of 20–90°, (**b**) the enlarged graphs in the range 27.5 < 2θ < 30.5 and 58.5 < 2θ < 60.5; for 4 mol% partially stabilized zirconia: (**c**) in the 2θ range of 20–90°, (**d**) the enlarged graphs in the range 27.5 < 2θ < 30.5 and 58.5 < 2θ < 60; and for 5 mol% partially stabilized zirconia: (**e**) in the 2θ range of 20–90°, (**f**) the enlarged graphs in the range 27.5 < 2θ < 30.5 and 58.5 < 2θ < 60.5. The control groups had tetragonal and cubic crystal phases. After sandblasting, the appearance of monoclinic phase peak (−111) at 2θ = 28.2° and rhombohedral phase peak (12 − 1) at 2θ = 29.88 were identified in three zirconia grades. m: monoclinic phase; t: tetragonal phase; c: cubic phase; r: rhombohedral phase.

**Figure 5 materials-14-05321-f005:**
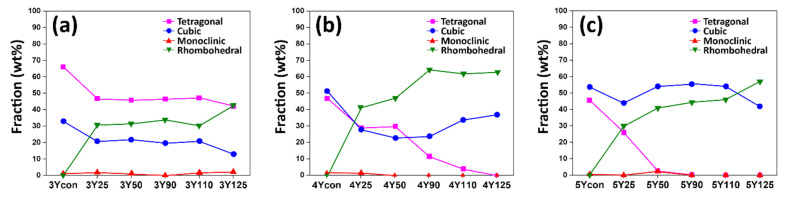
Values of the phase fraction (as weight percent) determined by Rietveld analysis of the XRD patterns for (**a**) 3Y subgroups, (**b**) 4Y subgroups, and (**c**) 5Y subgroups. The tetragonal phase contents decreased while the rhombohedral phase contents increased as the blasting particle size increased for all zirconia grades.

**Figure 6 materials-14-05321-f006:**
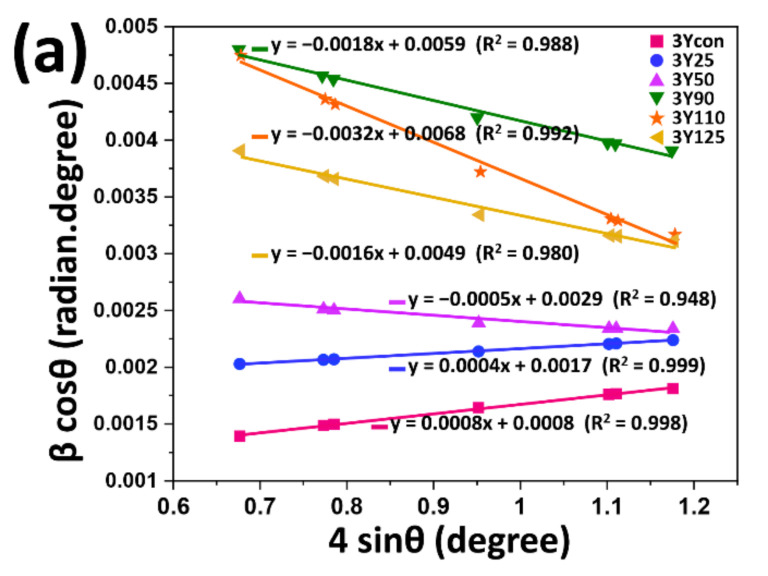
Williamson-Hall plots of ß cos θ against 4 sin θ calculated from XRD spectra for (**a**) 3Y subgroups, (**b**) 4Y subgroups, and (**c**) 5Y subgroups. From the linear fit to the data, 3Y110 showed the steepest negative slope while there were no lines with negative slopes for 4Y or 5Y subgroups.

**Figure 7 materials-14-05321-f007:**
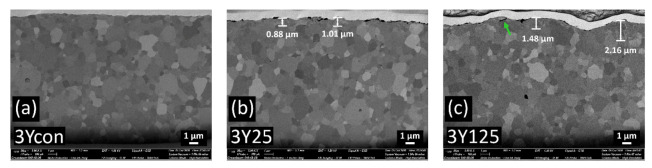
FIB cross-sections of (**a**) 3Ycon, (**b**) 3Y25, (**c**) 3Y125, (**d**) 3Y110; the thin boxes indicate the locations of details (**e**,**f**). (**e**) detail of the transformed zone of 3Y110, (**f**) detail of the untransformed zone of 3Y110. In detail (**e**), the transformed monoclinic grains with twinning are indicated by the blue arrows and the small pores are indicated by the yellow arrows. No microcracks are detected. In detail (**f**), there is no transformed grain. The small microcrack (shown in green arrow) is detected close to the surface in 3Y125.

**Figure 8 materials-14-05321-f008:**
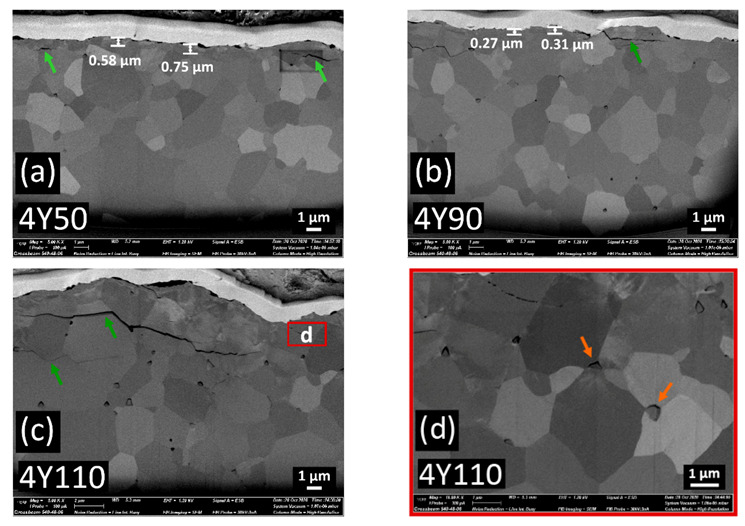
FIB cross-sections of (**a**) 4Y50, (**b**) 4Y90, (**c**) 4Y110; the thin box indicates the location of detail (**d**), (**d**) detail of the deformed zone of 4Y110. Microcracks (shown in green arrows) are present and are limited to the first 1 µm in 4Y50, while microcracks grow in the horizontal direction and are connected to the surface in 4Y90. 4Y110 shows the homogenous deformed layer instead of isolated transformed grains beneath the surface and lateral cracks are located up to 4.5 µm under the surface. The transverse extension of cracks could generate material removal (brittle fracture). In detail (**d**), the grains boundaries are destroyed and plastic deformations are detected. Alumina particle debris (shown in orange arrows) was deposited beneath the surface.

**Figure 9 materials-14-05321-f009:**
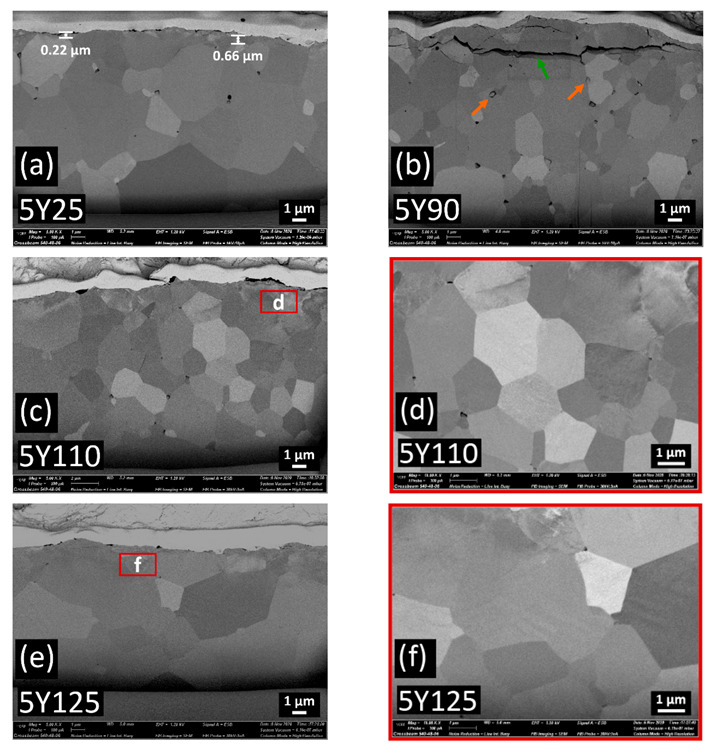
FIB cross-sections of (**a**) 5Y25, (**b**) 5Y90, (**c**) 5Y110; the thin box indicates the location of detail (**d**), (**d**) detail of the deformed zone of 5Y110, (**e**) 5Y125; the thin box indicates the location of detail (**f**), (**f**) detail of the deformed zone of 5Y125. Thin transformed layer is found below 0.7 µm under the surface in 5Y25, while lateral crack is connected to the surface without isolated transformed grain zone in 5Y90. Alumina particle debris (shown in orange arrows) was detected in 5Y90. In detail (**d**) and detail (**f**), homogeneous deformed zones with plastic deformation and surface melting are found. Abnormal grain growth is observed in 5Y125.

**Figure 10 materials-14-05321-f010:**
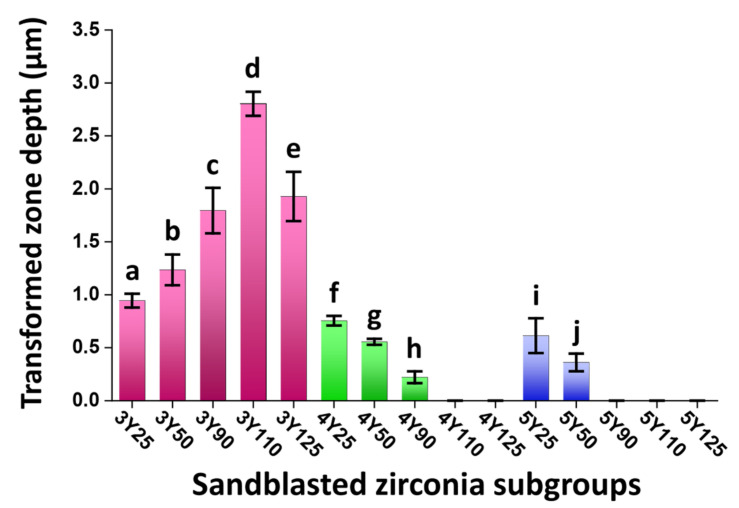
Means of the tetragonal to monoclinic phase transformed zone depth obtained from FIB/SEM images. A two-way ANOVA revealed that there was a statistically significant interaction between the effects of zirconia grade and the abrasive particle size on the transformed zone depth (*p* < 0.05). Error bars denote standard deviations around the means. Means with different lowercase letters show statistically significant differences within each zirconia grade based on a Tukey honestly significant difference multiple comparison test (*p* < 0.05). For 3Y subgroups, 3Y110 induced the deepest transformed layer up to 2.9 µm. For 4Y subgroups, 4Y25 induced the deepest transformed layer up to 0.8 µm. For 5Y subgroups, 5Y25 induced the deepest transformed layer up to 0.7 µm. No isolated transformed zone was found under the surface beyond 4Y110 for 4Y subgroups and beyond 5Y90 for 5Y subgroups.

**Figure 11 materials-14-05321-f011:**
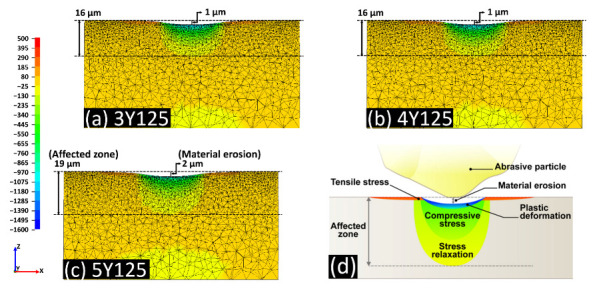
The hemispherical stress fields on loading within the model of (**a**): 3Y125; (**b**): 4Y125; (**c**): 5Y125; and (**d**): schematic view of the impact stress field. The compressive stress (negative) was concentrated at the impact zone and changed into tensile residual stress (positive) approaching the surface. 5Y125 produced deeper erosive cut compared to 3Y125 or 4Y125 with a depth of 2 µm. Below the material erosion, plastic deformation zone was developed. 5Y125 exhibited deeper stress relaxation zone with a depth of 19 µm compared to 3Y125 or 4Y125. MPS: maximum principal stress.

**Figure 12 materials-14-05321-f012:**
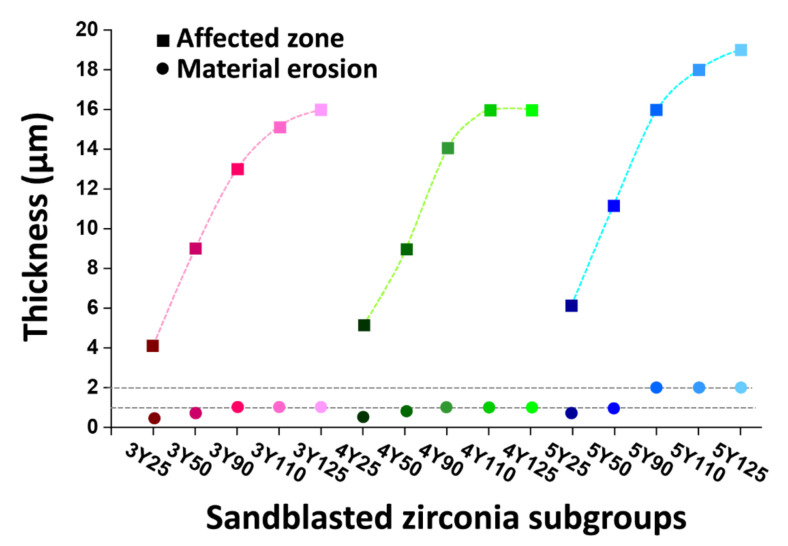
Thicknesses of the affected stress layers and the material cuts below the impact zones for the finite element modeling of all zirconia grades. 5Y subgroup models showed the deeper impact cuts at the contact zone compared to 3Y or 4Y subgroup models.

**Table 1 materials-14-05321-t001:** Material elastic properties of 3D-FEA models.

Material		Particle Size (µm)	Density (g/cm^3^)	Poisson’s Ratio	Young’s Modulus (GPa)	Flexural Strength (MPa)
Zirconia	ML (3Y-TZP)	0.52 ± 0.05	6.10 ^a^	0.30 ^a^	210 ^a^	800–900 ^a^
	STML (4Y-PSZ)	1.19 ± 0.20	6.10 ^a^	0.30 ^a^	210 ^a^	560–650 ^a^
	UTML (5Y-PSZ)	1.58 ± 0.17	6.10 ^a^	0.30 ^a^	210 ^a^	470–500 ^a^
Abrasive Particle	Al_2_O_3_	25, 50, 90, 110, 125	3.95 ^b^	0.22 ^b^	375 ^b^	379 ^b^

3Y-TZP: 3 mol% yttria-stabilized tetragonal zirconia polycrystal; 4Y-PSZ: 4 mol% partially stabilized zirconia; 5Y-PSZ: 5 mol% partially stabilized zirconia.^a^ Values reported by Kaizer et al. [25] ^b^ Values reported by Boniecki et al. [26].

**Table 2 materials-14-05321-t002:** Maximum principal stresses in the model designs of each zirconia subgroup. All values in megapascal (MPa).

Model Design		Maximum Principal Tensile Stress (Positive Values)	Maximum Principal Compressive Stress (Negative Values)
3Y	3Y25	416.06	1533.85
	3Y50	402.89	1542.26
	3Y90	480.76	1460.81
	3Y110	425.25	1416.71
	3Y125	397.41	1576.75
4Y	4Y25	412.96	1543.43
	4Y50	402.98	1554.53
	4Y90	479.02	1472.94
	4Y110	430.63	1423.44
	4Y125	399.69	1570.46
5Y	5Y25	402.63	1397.91
	5Y50	399.91	1408.50
	5Y90	397.71	1308.88
	5Y110	376.20	1240.51
	5Y125	336.36	1306.07

## Data Availability

The data presented in this study are available on request from the corresponding author.

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
