# Peer review of "Phase Transformations and Subsurface Changes in Three Dental Zirconia Grades after Sandblasting with Various Al2O3 Particle Sizes"

_materials, 2021, doi:10.3390/ma14185321_

Round 1

Reviewer 1 Report

The pre sent study aimed to evaluate the phase transformations and subsurface changes after sandblasting three different dental zirconia. The subject is interesting and can contribute to the scientific literature.

Abstract:

Correct the bold in the word Although;

Describe the Zirconia specimens dimensions;

Rewrite your conclusion focusing in the phase transformations instead bonding. The bond strength was not measured.

Introduction:

Please reference the following statement: “However, in terms of clinical durability, one of the major concerns associated with zirconia restorations would be their problematic bonding to resin cements due to their chemically inert nature and hardness.”;

Please introduce the relationship between sandblasting particle size and surface defects.

Methods:

How the sample size was determined?

Specify the ImageJ parameters of measurement.

Describe the FEM analysis type.

Improve the boundary conditions explanation for the finite element model.  

Results:

The region of material erosion in figure 11 has been calculated by adaptive mesh and element deletion or is it an overexpression of the results scale? Please improve it in the methods section.

Conclusion:

Please shorten it avoiding results repetition and the bonding statement. What are the clinical implications of your study?

Author Response

We, the authors, highly appreciate the detailed valuable comments on this manuscript.

The suggestions are quite helpful for us and we incorporate them in the revised paper.

The revision was listed below the comments and recommendations one by one.

================================================================

Response to Reviewer 1 Comments

The present study aimed to evaluate the phase transformations and subsurface changes after sandblasting three different dental zirconia. The subject is interesting and can contribute to the scientific literature.

Abstract:

Correct the bold in the word Although;

- We corrected the bold in the word “Although” in the revised manuscript. Thank you.

Describe the Zirconia specimen’s dimensions;

- We described the specimen’s dimension (14.0x14.0x1.0 mm3) in the revised manuscript.

Rewrite your conclusion focusing in the phase transformations instead bonding. The bond strength was not measured.

- We rewrote the conclusion in the revised manuscript: The recommended sandblasting particles were 110 µm for 3Y-TZP and 50 µm for 4Y-PSZ or 5Y-PSZ for compressive stress-induced phase transformations without significant subsurface damages.

Introduction:

Please reference the following statement: “However, in terms of clinical durability, one of the major concerns associated with zirconia restorations would be their problematic bonding to resin cements due to their chemically inert nature and hardness.”;

- We added the reference in the revised manuscript: However, in terms of clinical durability, one of the major concerns associated with zirconia restorations would be their problematic bonding to resin cements due to their chemically inert nature and hardness [1].

Please introduce the relationship between sandblasting particle size and surface defects.

- We introduced the relationship between sandblasting particle size and surface defects in the revised manuscript: Thus, the impact of hit can depend on the particle mass. Essentially, the bigger blasting particles caused morphological defects in the zirconia surfaces [8].

Methods:

How the sample size was determined?

- We prepared a total of 12 specimens per each grade then, specimens of each grade were divided into 6 groups (n = 2) according to the alumina abrasive particle size.

Specify the ImageJ parameters of measurement.

- We specified the ImageJ parameters of measurement in the revised manuscript: Ten measurements of the transformed zone depth on each FIB cross-section were carried out using ImageJ software (v1.53e, National Institute of Health, Bethesda, MD, USA) with the line measuring tool to measure a length distance over an image.

Describe the FEM analysis type.

- We described the FEM analysis type in the revised manuscript: After constructing the models, a linear elastic analysis under a dynamic load was performed. The indentation models assumed that the affected stress fields underneath the particle impact were hemispherical symmetries.

Improve the boundary conditions explanation for the finite element model.

- We explained the boundary conditions for the finite element model in the result section of the revised manuscript: The affected zone near the blasted surface was divided into four zones: (i) plastic deformation zone with a depth of a few micrometers with a certain degree of microstructural deformations below the material erosion; (ii) tensile stress zone in which the tensile stress was generated at the surface; (iii) compressive stress zone in which the residual compressive stress was induced beneath the particle impact; (iv) stress relaxation zone in which the residual stress was partially released.

Results:

The region of material erosion in figure 11 has been calculated by adaptive mesh and element deletion or is it an overexpression of the results scale? Please improve it in the methods section.

- We described how the region of material erosion has been calculated in the revised manuscript: The region of material erosion was calculated by adaptive mesh and element deletion [23].

Conclusion:

Please shorten it avoiding results repetition and the bonding statement. What are the clinical implications of your study?

We shorten the conclusion section in the revised manuscript:

We have evaluated the crystallographic and microstructural subsurface changes after alumina sandblasting with 5 different particle sizes in 3 different dental zirconia grades. Within the limitations of this in vitro study, the following conclusions were drawn:

  1. Although alumina sandblasting induced tetragonal to monoclinic phase transformation in the conventional zirconia, the phase transformation would depend on the amount of metastable tetragonal phase in highly translucent zirconia.
  2. When selecting the appropriate sandblasting protocols of highly translucent zirconia, care should be taken in order not to deteriorate the mechanical properties due to the high susceptibility to surface damage under applied stress.
  3. Within the range of treatment parameters investigated in this study, the recommended sandblasting particles were 110 µm for 3Y-TZP and 50 µm for 4Y-PSZ or 5Y-PSZ at a pressure of 0.2 MPa and a distance of 10 mm from the specimen’s surface for 10 s/cm2 to implement compressive stress-induced phase transformations without significant subsurface damages.

Reviewer 2 Report

This is a potential interesting manuscript dealing the phase transformation and subsurface changes in three dental zirconia grades after sandblasting with various Al2O3 particle size. Some minor revisions are suggested. (1) Figure 5, legend: should describe the content of (a) (b) and (c).  Similarly also the presentation of results in text (Figure 4 and 5, (a) (b) (c) ….etc.). (2) In Figure 10 and others, why choose to use two-way ANOVA for statistical analysis?

Author Response

We, the authors, highly appreciate the detailed valuable comments on this manuscript.

The suggestions are quite helpful for us and we incorporate them in the revised paper.

The revision was listed below the comments and recommendations one by one.

================================================================

Response to Reviewer 2 Comments

This is a potential interesting manuscript dealing the phase transformation and subsurface changes in three dental zirconia grades after sandblasting with various Al2O3 particle size.

Some minor revisions are suggested.

(1) Figure 5, legend: should describe the content of (a) (b) and (c). Similarly also the presentation of results in text (Figure 4 and 5, (a) (b) (c) ….etc.).

- We described the content of (a), (b), and (c) in Figure 5 in the revised manuscript: Figure 5. Values of the phase fraction (as weight percent) determined by Rietveld analysis of the XRD patterns for (a) 3Y subgroups, (b) 4Y subgroups, and (c) 5Y subgroups

- We described the content of Figure 4 and 5, (a) (b) (c) ….etc. in the text of the revised manuscript:

Figure 4 reveals the XRD patterns and the enlarged graphs in the 2θ angle range from 27.5° to 30.5° and in the 2θ angle range from 58.5° to 60.5° for all subgroups of each zirconia grade: (a) and (b) for 3Y subgroups; (c) and (d) for 4Y subgroups; and (e) and (f) for 5Y subgroups. Figure 5 demonstrated quantitative phase distributions obtained from Rietveld refinements for (a) 3Y subgroups, (b) 4Y subgroups, and (c) 5Y subgroups.

(2) In Figure 10 and others, why choose to use two-way ANOVA for statistical analysis?

- In this study, a two-way ANOVA test was used to determine the effect of two nominal predictor variables (zirconia grade: 3Y, 4Y, and 5Y; and abrasive particle size: 25 µm, 50 µm, 90 µm, 110 µm, and 125 µm) on a continuous outcome variable (transformed zone depth after sandblasting).

Reviewer 3 Report

Dear Author

The aim of this study was to compare phase transformations and subsurface changes after sandblasting with five different of alumina particle sizes in 3 different zirconia (3, 4, and 5 mol% yttria-stabilized zirconia; 3Y-TZP, 4Y-PSZ, and 5Y-PSZ). The idea of this manuscript is very clear and simple. Basically, I like this kind of mechanical test combine with in silico test.

First of all, manuscript style is not well managed. M and M is clear. The results are very clear, and the data is trustable. The discussion is logically related with the test method, but little bit shallow. Conclusions are basically base on the results, however effect of bond strength was not tested in this study, so that is not possible to conclude about “better bonding”.

My specific comments below.

Abstract: The aim in abstract is not same as in the main text.

Abstract: “The recommended sandblasting particles were 110 µm for 3Y-TZP and 50 µm for 4Y-PSZ or 5Y-PSZ for better bonding without a significant reduction in the mechanical strength.”. This manuscript was not done about bonding test. So this conclusion is not suitable.

Mnufacture name: Kuraray Noritake should be changed as Kuraray Noritake Dental.

0.2 MPa pressure is seems like low for Zirconia air abrasion. why this pressure value was used?

The importance of add the FEM test is not understandable. Please add more detailed discussions about relation between mechanical test and numerical test.  

Conclusion 4: “better bonding without” should be removed. Because this manuscript was not done the bonding test. It can not conclude.

In Figure 7 (c), picture quality is not acceptable. Please change this picture as high resolution.

Author Response

We, the authors, highly appreciate the detailed valuable comments on this manuscript.

The suggestions are quite helpful for us and we incorporate them in the revised paper.

The revision was listed below the comments and recommendations one by one.

================================================================

Response to Reviewer 3 Comments

Dear Author

The aim of this study was to compare phase transformations and subsurface changes after sandblasting with five different of alumina particle sizes in 3 different zirconia (3, 4, and 5 mol% yttria-stabilized zirconia; 3Y-TZP, 4Y-PSZ, and 5Y-PSZ). The idea of this manuscript is very clear and simple. Basically, I like this kind of mechanical test combine with in silico test.

First of all, manuscript style is not well managed. M and M is clear. The results are very clear, and the data is trustable. The discussion is logically related with the test method, but little bit shallow. Conclusions are basically base on the results, however effect of bond strength was not tested in this study, so that is not possible to conclude about “better bonding”.

My specific comments below.

Abstract: The aim in abstract is not same as in the main text.

- We rewrote the aim of this study in the main text of the revised manuscript: Therefore, the purpose of this study was to evaluate the phase transformations and subsurface changes after sandblasting with various Al2O3 particles in 3 different dental zirconia (3Y-TZP, 4Y-PSZ, and 5Y-PSZ), which would be crucial to predict the long-term mechanical behavior of dental zirconia systems.

Abstract: “The recommended sandblasting particles were 110 µm for 3Y-TZP and 50 µm for 4Y-PSZ or 5Y-PSZ for better bonding without a significant reduction in the mechanical strength.”. This manuscript was not done about bonding test. So this conclusion is not suitable.

- We rewrote the conclusion in the revised manuscript: The recommended sandblasting particles were 110 µm for 3Y-TZP and 50 µm for 4Y-PSZ or 5Y-PSZ for compressive stress-induced phase transformations without significant subsurface damages.

Mnufacture name: Kuraray Noritake should be changed as Kuraray Noritake Dental.

- We changed it to “Kuraray Noritake Dental” in the revised manuscript.

0.2 MPa pressure is seems like low for Zirconia air abrasion. why this pressure value was used?

- Most dental studies on the sandblasting effect for zirconia have been conducted under 0.1-0.4 MPa. In the Introduction section, “Those studies included alumina abrasive particles (grain size: 25-125 µm) and compressed air (pressure: 0.1-0.4 MPa) which are mixed to form high-speed abrasive flow after passing through the nozzle.” Chintapallie et al. (Reference #15) reported that mild sandblasting was more beneficial than the harsh one since the former induced limited damage. The biaxial flexural test showed improved mechanical strengths for the samples after sandblasting at 0.20–0.35 MPa, with the maximum strength at 0.25 MPa. Sandblasting at 0.40 MPa decreased the strength as compared with 0.25 MPa in Okada et al.’s study (Okada, M., Taketa, H., Torii, Y., Irie, M., Matsumoto, T. Optimal sandblasting conditions for conventional-type yttria-stabilized tetragonal zirconia polycrystals. Dent. Mater. 2019;35:169–175. doi: 10.1016/j.dental.2018.11.009.). Therefore, we used 0.2MPa pressure for zirconia air abrasion in this study.

The importance of add the FEM test is not understandable. Please add more detailed discussions about relation between mechanical test and numerical test.

- We added more detailed discussions about relation between mechanical test and numerical test in the revised manuscript: We performed sandblasting on the zirconia surfaces as a mechanical test to identify the crystallographic and morphological changes after sandblasting in three different zirconia grades. Additionally, the numerical analysis by means of the finite element method was used to determine theoretical stress distributions as well as to understand the empirical data from the mechanical testing.

Conclusion 4: “better bonding without” should be removed. Because this manuscript was not done the bonding test. It can not conclude.

- We rewrote the conclusion: Within the range of treatment parameters investigated in this study, the recommended sandblasting particles were 110 µm for 3Y-TZP and 50 µm for 4Y-PSZ or 5Y-PSZ at a pressure of 0.2 MPa and a distance of 10 mm from the specimen’s surface for 10 s/cm2 to implement compressive stress-induced phase transformations without significant subsurface damages.

In Figure 7 (c), picture quality is not acceptable. Please change this picture as high resolution.

- We changed Figure 7(c) as high resolution in the revised manuscript.
